# Re-Evaluating Human Cytomegalovirus Vaccine Design: Prediction of T Cell Epitopes

**DOI:** 10.3390/vaccines11111629

**Published:** 2023-10-24

**Authors:** Peter A. Barry, Smita S. Iyer, Laura Gibson

**Affiliations:** 1Department of Pathology and Laboratory Medicine, Center for Immunology and Infectious Diseases, University of California Davis School of Medicine, Sacramento, CA 95817, USA; pabarry@ucdavis.edu; 2California National Primate Research Center, University of California, Davis, CA 95616, USA; 3Department of Pathology, University of Pittsburgh, Pittsburgh, PA 15261, USA; ssi13@pitt.edu; 4Departments of Medicine and of Pediatrics, Infectious Diseases and Immunology, University of Massachusetts Chan Medical School, Worcester, MA 01655, USA

**Keywords:** cytomegalovirus, vaccine, bioinformatics, unconventional T cell antigen candidates

## Abstract

HCMV vaccine development has traditionally focused on viral antigens identified as key targets of neutralizing antibody (NAb) and/or T cell responses in healthy adults with chronic HCMV infection, such as glycoprotein B (gB), the glycoprotein H-anchored pentamer complex (PC), and the unique long 83 (UL83)-encoded phosphoprotein 65 (pp65). However, the protracted absence of a licensed HCMV vaccine that reduces the risk of infection in pregnancy regardless of serostatus warrants a systematic reassessment of assumptions informing vaccine design. To illustrate this imperative, we considered the hypothesis that HCMV proteins *infrequently* detected as targets of T cell responses may contain important vaccine antigens. Using an extant dataset from a T cell profiling study, we tested whether HCMV proteins recognized by only a small minority of participants encompass any T cell epitopes. Our analyses demonstrate a prominent skewing of T cell responses away from most viral proteins—although they contain robust predicted CD8 T cell epitopes—in favor of a more restricted set of proteins. Our findings raise the possibility that HCMV may benefit from evading the T cell recognition of certain key proteins and that, contrary to current vaccine design approaches, including them as vaccine antigens could effectively take advantage of this vulnerability.

## 1. Introduction

A 1971 paper by James Hanshaw chronicling a 15-year perspective of congenital human cytomegalovirus (cHCMV) concluded that “If these preliminary assessments [of societal burden of cHCMV sequelae] prove to be even near correct, any thoughtful program designed at prevention or treatment is warranted” [1]. Dr. Hanshaw’s conclusion was based on his estimation of ~5000 additional children with neurological sequelae of cHCMV infection born each year in the United States.

This early assessment of the societal impact of cHCMV was prescient. HCMV is the most frequent congenital infection, with an estimated global birth prevalence of 0.64 [2] and reported locale-specific frequencies ranging from 0.3 to 5.4% [3]. A recent study estimated that >700,000 infants are born with HCMV each year across Africa, Latin America, and Asia, regions where HCMV seroprevalence is high by the time children reach puberty [4]. Based on approximately 135,300,000 global births in 2019 [5] and assuming a global cHCMV birth prevalence of 0.64%, the annual number of congenital infections could be > 860,000 [2]. Moreover, cHCMV has revealed profound racial and income disparities from which these communities bear higher burdens of cHCMV disease [6,7,8,9,10,11,12].

The mucosal acquisition of live virions after contact with infectious body fluids is the typical mode of HCMV transmission between individuals, which for pregnant individuals is often contact with young children shedding HCMV. Primary infection is characterized by organ dissemination, a relatively rapid clearance of viremia, and prolonged, although variable, duration and magnitude of viral shedding in fluids of the salivary glands, genitourinary tract, testes, mammary gland, and cervix [13,14,15,16,17,18,19,20,21,22,23,24,25]. In contrast, occult non-primary infection involves the reactivation of a latent virus or re-infection with a new viral population, although the available evidence favors the latter as having the higher risk of fetal transmission [26]. While only 14% of women of reproductive age globally are HCMV seronegative and at risk for primary infection [27], vaccine development efforts have focused almost exclusively on this population [28], thus leaving the HCMV seropositive majority with no apparent benefit. Nevertheless, an estimated 75% of infected infants are born to HCMV-seropositive women in the context of non-primary HCMV infection [3,27,29,30,31]. Having a major impact on our understanding of cHCMV epidemiology, more recent studies have found that the frequency of cHCMV infection in a population is directly related to maternal HCMV seroprevalence [27]. A transformational insight from these studies is that all pregnant women are at risk of fetal transmission regardless of preconceptional HCMV serostatus [7,8,9,10,32]. Without a measurable biomarker such as seroconversion to confirm primary infection [33], the critical aspects of non-primary infection, such as the protective role of prior immunity or rates of reinfection or fetal transmission, have been estimated but remain unknown [34,35].

cHCMV has long been recognized as a threat to fetal growth and development [1]. In utero and postnatal sequelae occur in about 10–15% of infected neonates and can range from subtle to severe; almost 20% of all infected infants have permanent disabilities regardless of disease severity at birth [29,31,36,37]. Accruing costs include the physical and emotional impacts on infected individuals and their families, the educational challenges imposed by neurodevelopmental deficits, and the financial liabilities borne by families and healthcare systems [38,39,40]. Accordingly, clinical and societal imperatives to reduce our collective burden of cHCMV compel a vaccine strategy that protects all pregnant individuals from HCMV infection and/or fetal transmission. However, replicating the features of protective host immunity in effect at any given point in HCMV natural history—viral shedding in body fluids, spread via mucosal surfaces, primary or non-primary infection during pregnancy, placental infection, or fetal transmission—remains a major challenge for any prophylactic or therapeutic intervention. In particular, the HCMV vaccine design may need to account for viral genomic mutation and immune escape, viral populations adapted to distinct tissue compartments (e.g., salivary glands or kidneys), or the evolution of immune specificities after initial and potentially multiple reinfection episodes over a lifetime.

HCMV is a consummate manipulator of the host immune system [41,42,43,44,45]. Protracted co-evolution with the vertebrate immune system over >400 million years has enabled HCMV and the other members of the *Herpesviridae* family [46] to establish and maintain lifelong persistence, undergo periodic reactivation that allows transmission, and re-infect through repeated mucosal exposures to antigenic variants [3,42,47]. The relationship between HCMV and its host favors mutual survival, such that memory immune responses measured during chronic infection provide a cumulative signature of previous virus–host interactions. While the antigen specificities of these responses can be measured, evidence linking specific viral targets with the control of HCMV replication or disease is lacking. Based on antigens recognized by healthy adults with chronic infections, vaccines to block primary HCMV have targeted proteins such as glycoprotein B (gB), the glycoprotein H (gH)-anchored pentamer complex (PC), and the unique long 83 (UL83)-encoded phosphoprotein 65 (pp65) [3,28,29,48,49,50,51,52,53]. Yet despite extensive efforts over more than 45 years since Elek and Stern identified de novo antiviral immune responses following inoculation of seronegative volunteers with live attenuated HCMV AD169 [54] and partial but incomplete success in clinical trials enrolling a variety of seronegative populations [28], no HCMV vaccine has been licensed. As a result, heretofore established precepts should be systematically re-evaluated and alternative assumptions and strategies considered in HCMV vaccine development.

To illustrate this imperative, we re-examined the question of optimal HCMV epitopes for T cells by considering the hypothesis that HCMV proteins *infrequently* recognized by T cells may contain targets that favor protection from transmission, infection, or disease. We leveraged computational tools for an in silico rather than experimental approach to T cell antigen discovery across the whole viral proteome. We envisioned a clinical paradigm whereby a prophylactic and/or therapeutic HCMV vaccine shifts the host–pathogen dynamic toward human advantage in such populations as young children with prolonged viral shedding, pregnant individuals regardless of HCMV history, or infants with cHCMV infection. This study highlights only one of many possible non-traditional analyses that could accelerate the development of a vaccine intended to reduce the incidence of cHCMV infection.

## 2. Methods

### 2.1. Overview

Our starting point was a seminal study interrogating T cell responses to the HCMV proteome in healthy immunocompetent adults with chronic HCMV infection [55]. The study was so prodigious in scope and detail that this unique body of work may never be repeated, so the dataset was explored for additional clues about anti-viral T cell targets that might catalyze novel HCMV vaccine design strategies. The original data were re-analyzed using a vantage point gained from 18 subsequent years of HCMV natural history and genomic diversity investigations and new computational algorithms for predicting T cell epitopes. 

In that report and its Appendix A [55], ss the study population (33 seropositive and 10 seronegative healthy adults) was based on broad HLA and ethnic diversity (Appendix A of [55]). No other parameters, such as HCMV PCR in blood, clinical symptoms, or viral shedding in bodily fluids, were provided. The original experiments used libraries of 15-mer peptides overlapping by 10 amino acids, corresponding to the annotated sequences of the laboratory-adapted AD169 strain (13,687 peptides from 191 proteins) and of either the Towne or Toledo strains (968 peptides from 22 proteins) to map CD4 and CD8 T cell responses (Appendix A of [55]; the paper states that 213 proteins were analyzed, but Appendix A lists 214 proteins). PBMCs stimulated with these peptide pools were gated for CD4^+^/interferon (IFN)-γ^+^/CD69^+^ or CD8^+^/IFN-γ^+^/CD69^+^ T cells and for memory T cell subsets (CD4^+^/CD45RO^+^/CD27^+^ or CD8^+^/CD95^+^) (results summarized in Appendix A of [55]).

### 2.2. Creating a Revised Set of HCMV Proteins

Compared to unpassaged clinical isolates, the original annotation of AD169 included multiple errors in the protein coding content due to genetic mutations accumulated during an extensive passage in cultured cells [56]. For the current study, a “Revised Set” of HCMV proteins was chosen by removing proteins in the original report [55] that were (a) duplicate sequences, (b) less than full-length proteins, (c) found in laboratory or other strains but not also in clinical isolates, and (d) originally described as distinct proteins but on review were different exons of the same protein, which were instead combined and analyzed as a single protein.

### 2.3. T Cell Responses to the Revised Set of HCMV Proteins 

In the original study, the HCMV proteins (*n* = 214) were stratified by CD4 or CD8 T cell reactivities (detectable IFN-γ^+^ production in response to HCMV peptide pool stimulation), representing either <1% or ≥1% of the memory T cell compartment. Since the virological and/or immunological rationale of this stratification was not described, the data for this study were categorized not by the relative magnitude of T cell reactivity but instead by the number of participants with detectable CD4 or CD8 T cell responses for each protein in the Revised Set. Based on our hypothesis, infrequent recognition by study participants was arbitrarily defined as “no” (0) or “few” (1–4) T cell responders and compared to “some” (4–16) or “many” (≥17) responders. Excluding all others, proteins with no/few (0–4) CD4 T cell and no/few (0–4) CD8 T cell responders were included in the “Test Set” for subsequent analyses.

### 2.4. Protein Sequence Identity

Assuming that strongly conserved HCMV proteins would stimulate T cell responses against the largest number of circulating viral populations, each protein in the Test Set was evaluated for genomic conservation in the GenBank database using the BLASTP (protein-protein BLAST) search algorithm. Variables included “non-redundant protein sequences” for Database and “Cytomegalovirus (taxid:10358)” for Organism, and the search for each protein was initiated using its GenBank Access number provided (Appendix A of [55]). The mean, median, and range of the amino acid sequence identity were only calculated from the BLASTP results, showing 100% protein sequence coverage. An arbitrary threshold of ≥98% protein sequence identity was used to include HCMV proteins in the “Identity Set” for the next step of the analysis. 

### 2.5. MHC Class I-Peptide Binding Affinity

The ideal HCMV vaccine would maximize the number of people that respond to vaccine antigen(s). Therefore, the HCMV proteins in the Identity Set were further examined in silico for peptide-MHC (pMHC) binding affinity to class I alleles common in the global human population. Our analysis focused only on CD8 T cell epitope prediction, primarily to test our hypothesis with a limited number of HCMV proteins, not to diminish the roles of CD4 T cells, B cells, NK cells, or any other aspect of innate or adaptive immunity to HCMV infection. Predictions were made using the TepiTool analytics modality [57] of the Immune Epitope Database (IEDB) (http://tools.iedb.org/tepitool/). Each protein in the Identity Set was investigated for peptides that are predicted to bind with high affinity to most (arbitrarily defined as ~75% or >21) of the 27 MHC class I alleles in the tool. The following variables were selected: (1) a single HCMV protein was entered for each analysis, (2) human as the host species and class I as the allele class, (3) specific MHC alleles as the “Use panel of 27 most frequent A & B alleles” (A*01:01, A*02:01, A*02:03, A*02:06, A*03:01, A*11:01, A*23:01, A*24:02, A*26:01, A*30:01, A*30:02, A*31:01, A*32:01, A*33:01, A*68:01, A*68:02, B*07:02, B*08:01, B*15:01, B*35:01, B*40:01, B*44:02, B*44:03, B*51:01, B*53:01, B*57:01, B*58:01), (4) peptide length as “Apply default settings for moderate number of peptides” (8–11 amino acids), (5) prediction method as “IEDB recommended”, and (6) predicted peptides as “Select peptides based on predicted IC50”. The default “IEDB recommended” uses the best prediction method based on the available data for each MHC allele. For MHC class I binding affinity, the Consensus method (combination of artificial neural network, stabilized matrix method, and CombLib) is typically used, or NetMHCpan if no allele data are available. A threshold of IC50 ≤ 500 nM is generally regarded as an appropriate pMHC binding affinity to gauge immunogenicity for CD8 T cells [58,59,60]. However, considering previous studies showing that most (60–90%) immunogenic peptides have MHC binding affinities ≤ 50 nM [59] and that higher affinity interactions are associated with greater immune responsiveness [58], a more stringent threshold of IC50 ≤ 50 nM was used to identify peptides most likely to bind MHC class I alleles in vivo. This analysis returned proteins in the “Affinity-HLA Set” expected to generate the highest possible number of worldwide CD8 T cell responders. 

### 2.6. Immunogenicity Prediction 

The optimal HCMV vaccine would also be maximally immunogenic for protective T cell responses in both primary and non-primary infections. Therefore, peptides in the Affinity-HLA Set were further analyzed using IEDB “T Cell Epitopes—Immunogenicity Prediction” (http://tools.iedb.org/immunogenicity/). This IEDB tool predicts the relative immunogenicity of the pMHC complex for CD8 T cells based on amino acid properties and positions [61]. Of note, the algorithm has been validated only for 9-mers bound to MHC class I molecules, but it compensates for the additional amino acids of longer peptides in terms of weighting and masking. The tool does not predict immunogenicity for CD4 T cells. 

To avoid skewing the aggregate immunogenicity scores, duplicate entries for peptides that bind multiple HLA alleles were first removed from the Affinity-HLA Set. The remaining unique peptide sequences were then submitted to the immunogenicity tool with masked positions set to default. The output was a list of predicted epitopes ranked by probability of eliciting a CD8 T cell response. Based on scores for component peptides, a median immunogenicity score was calculated for each protein and compared to that of UL83, the prototypic immunodominant HCMV protein for CD8 T cell responses. Proteins with a median score significantly below UL83 (2-tailed Mann-Whitney, GraphPad Prism, version 9) were excluded to yield the “Final Set” of HCMV epitopes for CD8 T cells.

## 3. Results 

### 3.1. Patterns of T Cell Responses to HCMV Proteins

Based on the original set of HCMV proteins (*n* = 214) used by Sylwester et al. [55], a Revised Set (*n* = 148 or 69.2% of the original set) was generated by excluding duplicate (*n* = 3), non-full-length (*n* = 12), and “AD169-only” (*n* = 48) proteins, and by combining separate exons of the same protein into a single re-annotated protein (*n* = 3) (Figure 1; Table 1; Appendix A). Proteins found only in AD169 and not also in clinical isolates were excluded because an analysis based solely on an extensively passaged laboratory strain may not have accurately reflected the T cell responses of study participants infected with wild-type variants.

Whether the original [55] or the Revised Set was examined, the results highlight a marked skewing of HCMV-specific CD4 or CD8 T cell responses in favor of certain proteins and away from those with no/few (0–4) responders (Figure 2; Appendix A). Of the 148 HCMV proteins in the Revised Set, 46 (31.1%) had no CD4 and 56 (37.8%) had no CD8 T cell responders, respectively, and 26 (17.6%) had neither CD4 nor CD8 T cell responders. Similarly, 74 (50.0%) had few CD4 and 67 (45.3%) had few CD8 T cell responders, respectively, and 38 (25.7%) had few CD4 and CD8 T cell responders (Figure 2; Appendix A). In marked contrast, a total of only six (4.1%) individual proteins had many CD4 (UL55, UL83, UL86, and UL99) or many CD8 (UL48, UL83, and UL123) responders; UL83 was the only protein recognized by both the T cell subsets (Figure 2). Taken together, most (109 or 73.6%) of the unique HCMV proteins in the Revised Set stimulated T cells in ≤four (≤12.1%) of the study participants, comparable to the original protein set (Figure 1C of [55]). This prominent skewing implies that a reciprocal small fraction (39 or 26.4%) of the HCMV proteins contained T cell epitopes that are commonly reported in healthy HCMV-infected adults [55].

Further, when the T cell responses were tabulated for each participant (Figure 3; Appendix A), no apparent patterns were identified in terms of (a) the number of proteins recognized by either the CD4 (median 11; range 2–35) or CD8 (median = 8; range 1–30) T cells or by both the CD4 and CD8 T cells (median 21; range 6–60), and (b) the proportion of participants for whom the responses were primarily CD4 (*n* = 21 or 67%) or CD8 (*n* = 11 or 33%) T cells. These results imply that the course of virus–host interactions from acute to persistent infection is different for each individual. This observation is highlighted in Figure 3 comparing (a) Participants 1 (P1) and 15 (P15), in which the T cell response patterns are predominantly either CD8 or CD4, respectively, or (b) P32 and P33, in which the total number of proteins eliciting T cell reactivities is quite discordant.

In contrast to the variability of the T cell response profiles for individual participants, a pattern in the relationship between the numbers of CD4 and CD8 T cell responders was observed (Figure 4). The proteins that stimulated CD8 T cells in no (0), few (1–4), or some/many (>4) participants tended to cluster with those that stimulated the same range of CD4 T cells. These results again suggest that only a restricted subset of proteins is detectable in the context of MHC class I and/or class II during acute/persistent HCMV infection.

### 3.2. Amino Acid Sequence Identity in HCMV Proteins with No or Few T Cell Responders

To focus the analysis on HCMV proteins with no/few (0–4) T cell responders, the proteins in the Revised Set (*n* = 148) with >4 CD4 and/or >4 CD8 T cell responders (*n* = 39 or 26.4%) were removed to generate the “Test Set” (*n* = 109 or 73.6%) (Figure 1; Appendix A). Since the breadth of the memory T cell responses against wild-type HCMV may not be fully stimulated by AD169-derived peptides [55], thus leading to no or few identified T cell responders, the proteins in the Test Set were normalized for amino acid sequence identity. The median identity was calculated for each protein using BLASTP (Appendix A). An arbitrary identity threshold of ≥98% was then applied to exclude the more heterogeneous proteins, which yielded an “Identity Set” of 61 proteins (56.0% of Test Set; 41.2% of Revised Set; Figure 1; Appendix A). With more than half of the proteins in the Test Set exhibiting nearly identical sequences, this BLASTP analysis shows that extensive sequence variation between the AD169-derived and wild-type HCMV peptides was not likely a reason why most of the HCMV proteins had no/few T cell responders.

### 3.3. Predicted Epitopes in Conserved HCMV Proteins with No or Few T Cell Responders

Although the original study population was selected for broad HLA and ethnic diversity [55], the no/few T cell responders to the majority of the HCMV proteins could reflect the limited peptide binding capability of the actual HLA repertoire represented by the study cohort. Therefore, the proteins in the Identity Set (*n* = 61) were further analyzed to find potential epitopes based on pMHC binding affinity. 

Overall, 12 of 15 (80.0%) HLA-A and 8 of 26 (30.1%) HLA-B alleles of the cohort were represented in the panel of 27 alleles in the TepiTool analysis. Every participant had at least two alleles in the panel, except P24 with only one. The analysis revealed 35 (57.4%) proteins predicted to contain peptides that bind with high affinity (IC50 ≤ 50 nM) to most (>75%) of the HLA alleles in the panel (Figure 1; Appendix A). This “Affinity-HLA Set” (*n* = 35) represents 32.1% of the Test Set (*n* = 109) and 23.6% of the starting Revised Set (*n* = 148). As expected, based on the immunodominance of UL48, UL83, and UL123 for the study cohort, the peptides within these proteins were predicted to bind with high affinity (≤50 nM) to 26 (96.3%), 22 (81.5%), and 21 (77.8%), respectively, of the panel alleles. These data suggest that limited high-affinity HLA binding is not an explanation for most of the HCMV proteins stimulating no/few T cell responders. 

### 3.4. Predicted Immunogenicity of Peptides in the Affinity-HLA Set 

Another possible reason for the limited responders is that, despite the proteins being highly conserved and containing multiple high-affinity MHC binding peptides, the pMHC complexes generated may not have sufficiently stimulated memory T cells in that particular study cohort. Proteins in the Affinity-HLA Set (*n* = 35) were therefore analyzed using the “T Cell Epitopes—Immunogenicity Prediction” tool of the IEDB to assess their potential to stimulate CD8 T cells. The output data were compared to the immunogenicity score of UL83, the prototypic immunodominant HCMV protein (Figure 1 and Figure 5; Appendix A). 

The proteins with median scores significantly lower than UL83 (*n* = 11) were removed to yield the “Final Set” of HCMV proteins (*n* = 24; Table 1). These data suggest that the low immunogenic potential of pMHC complexes does not explain the limited number of CD8 T cell responders to these particular 24 HCMV proteins. 

In summary, most of the HCMV proteins (*n* = 109, or 73.6% of the Revised Set) were recognized by the T cells of ≤4 (12.1%) study participants, yet at least 24 of them (Table 1 Final Set; 16.2% of the starting Revised Set) were predicted to contain CD8 T cell epitopes that have conserved amino acid sequences, to bind with high affinity to HLA alleles expressed by the cohort, and to have high immunogenic potential. Finding no apparent constraints at these key points in T cell stimulation, our analysis supports the hypothesis that HCMV proteins that are rarely detected as T cell antigens may contain valuable targets for these responses.

## 4. Discussion

Given the breadth and complexity of HCMV immune evasion tactics and limited progress in licensing a HCMV vaccine, we considered the hypothesis that HCMV proteins that are infrequently detected ex vivo as T cell targets may in fact contain viral epitopes that could generate T cell responses. We sought to illustrate the use of T cell epitope prediction tools for HCMV antigen discovery and to highlight that these and other preclinical computational methods could serve as a new prism in HCMV vaccine design. Toward these ends, our study reanalyzed an invaluable dataset characterizing T cell responses to the HCMV proteome in 33 healthy adults with chronic infection [55]. This analysis revealed that, despite being recognized by only a few or even none of the participants, most HCMV proteins contain peptides that are relatively likely to elicit robust HCMV-specific T cell responses based on the prediction parameters. 

Reverse vaccinology to leverage genomic data [62,63] is an exciting new process that may inform ex vivo experiments and increase the efficiency and cost-effectiveness of vaccine development, particularly for intractable challenges like HCMV. A few recent studies have used in silico immuno-informatics approaches to design theoretical HCMV subunit vaccines [64,65,66,67]. Among the first, Quizno et al. [66] identified HCMV epitopes for CD4 and CD8 T cells detected in humans and reported in the IEDB, using several methods to predict linear HCMV epitopes for B cells. They applied computational algorithms to filter epitopes by maximal amino acid sequence conservation (to mitigate potential genetic drift or immune escape), MHC-peptide binding affinity (for efficient MHC presentation), HLA allele population coverage (for global applications), and minimal human protein cross-reactivity (to avoid autoimmunity), yielding a final set of 15 peptides derived from four HCMV proteins. Other computational vaccine studies used alternative methods, including molecular docking and immunogenicity simulation, prediction of non-allergenicity, physicochemical properties of peptides, expression feasibility, and tertiary structure modeling of the vaccine construct [64,65,67]. By extension, comparable in silico approaches are being evaluated for *Mycobacterium tuberculosis*, *Plasmodia* species, and seasonal influenza variants [68,69,70,71]. In a recent novel approach targeting another *Herpesviridae* pathogen, a multivalent Epstein–Barr Virus (EBV) vaccine was constructed to include multiple humoral and CD8 T cell antigens of both lytic and latent proteins presented by common HLA class I alleles to enable global utility [72]. 

However, computational studies of HCMV epitopes tend to focus on a few proteins from which B and T cell epitopes have been reported primarily in healthy adults with chronic infection. Yet, searching for immunogenic peptides arising from a small fraction of >200 viral open reading frames (ORFs) likely has low sensitivity for identifying epitopes that control HCMV primary or re-infection in young children or in people with limited immune capacity. Like our study, a recent comprehensive analysis challenges the assumption that a limited number of “immunodominant” HCMV proteins is sufficient for an immune-based intervention to elicit protection from HCMV infection or disease [73]. To discover new HCMV epitopes from a proteome-wide perspective, Dhanwani et al. [73] tested a library of 2593 15-mer peptides predicted in part by IEDB to stimulate CD4 T cells in a screening cohort of 19 healthy adults, deconvoluted the top 10 reactive peptide pools (~11% of the total 89 pools tested), identified 235 epitopes (all novel) covering 100 ORFs (93 novel), and tested these peptides in a validation cohort of 20 individuals (10 HCMV seropositive). Of note, they highlighted the role of bioinformatics tools in reducing the number of peptides screened, thus demonstrating the efficiency and cost-effectiveness of this approach. Moreover, the top 10 pools accounted for ~90% of the responses detected in each participant, suggesting, like Sylwester et al. [55], that HCMV-specific memory T cells recognize a restricted set of viral peptides that varies by individual and raising, as in our study, the question of why the peptides in the remaining 79 pools were recognized by only a small subset of participants. 

A myriad of factors could explain the limited recognition of certain HCMV proteins by a small cohort of healthy adults with past HCMV infection. For example, T cells in the periphery represent only ~2% of all body lymphocytes [74,75], but since HCMV disseminates to most organs during primary infection, quantifying T cells in the blood may not offer a representative sample of all the within-host HCMV-specific T cells. Similarly, identifying HCMV-specific T cells using a functional marker such as IFN-γ could miss cells that have an alternative functional response(s). Our analysis focused on other possible factors operating on the original dataset used by Sylwester et al.: (a) hypervariable amino acid sequences such that peptide pools did not stimulate memory T cells previously generated against divergent viral populations, (b) peptides that did not bind with high enough affinity to the specific MHC class I or II molecules of some participants, and (c) pMHC complexes that were not sufficiently immunogenic to stimulate a detectable T cell response. 

Nevertheless, the capacity of a few computational models to forecast complex virus–host interactions is limited, especially when the performance of the models varies when compared using a standardized validation dataset [76]. T cell epitope prediction tools can differ by developer, number, and type of simulated parameters (e.g., peptide-MHC binding affinity or antigen processing), input HLA alleles, computational methods, training datasets, and other variables that affect performance. For example, TepiTool was only validated for 9-mer peptides [57], and the IEDB Immunogenicity model used relatively small training dataset that did not account for non-linear factors or position-specific amino acid enrichment, which can affect CD8 T cell stimulation [61]. However, they each have specific advantages, such as the “IEDB recommended” default in TepiTool that uses the optimal method based on regular data benchmarking of new and existing algorithms or the immunogenicity tool predictions based on amino acid properties, a parameter less commonly used by other algorithms. Several tools likely could have served our primary aim to emphasize the possibility that HCMV encodes many proteins that should be immunogenic but are not and to contemplate the potential impact and mechanisms for this observation.

Like HCMV, *Mycobacterium tuberculosis* (MTb) has co-evolved with humans for millions of years, leading to a dynamic of host survival and persistent bacterial infection [77,78,79]. A preponderance of strongly conserved T cell targets has been identified [79], suggesting a pathogen-mediated advantage of skewing responses to these proteins [78]. In one study, the authors note that “*One potential explanation* [for minimal protein escape variants] *is that during coevolution with humans,* [MTb] *has derived a net evolutionary benefit from T cell recognition, despite the within-host cost that T cell responses impose on the bacteria in the majority of infected individuals*” [79]. Similarly, *Plasmodium* species induce only partial immunity despite frequent reinfection in malaria-endemic areas, thus tolerating the cost of immune recognition while allowing individuals with chronic asymptomatic infection to survive and serve as reservoirs for transmission [80]. An analogous context is the rhesus macaque model of HCMV pathogenesis and persistence. The rhesus CMV (RhCMV) protein ortholog of HCMV pp65 (rh112/RhUL83B) is strongly conserved among RhCMV strains, ranging from 99.6 to 100% identical at the protein level [81,82,83]. The rh112/RhUL83B protein is an immunodominant target of T cell responses [84], yet, it is non-essential for viral replication in vitro. The inoculation of RhCMV-uninfected rhesus macaques with a rh112/RhUL83B-deleted variant leads to orders of magnitude greater replication in vivo than the parental construct, suggesting that the protein has a viral fitness cost rather than benefit [85]. This finding prompts the question of why expression and genomic sequence conservation would be favored for a protein that consistently stimulates anti-viral T cells unless they yielded a net survival advantage. Considering this question for HCMV, we speculate that HCMV mechanistically diverts T cells from proteins that favor viral persistence toward alternative “decoy” proteins (as suggested in [86]) that are relatively dispensable (e.g., UL83). This type of immune evasion could explain not only the net benefit of sequence conservation for the latter despite intense T cell selective pressure but also the infrequent T cell recognition of otherwise immunogenic but essential viral proteins. The clinical implication is that some ex vivo human studies may detect common yet non-protective T cell responses to viral antigens on which HCMV vaccine design is based.

This potential T cell evasion tactic presents both an enigma and an opportunity for reducing cHCMV disease burden. The enigma stems from the need to resolve the viral mechanisms that alter processing, presentation, binding to MHC or T cell receptors, or other pathways of adaptive immunity before manipulating the immunogenicity of “non-dominant” viral proteins during vaccine development. On the other hand, an opportunity lies in the premise that T cell-based interventions targeting proteins such as those in our Final Set may be exceptionally effective. While limited, our proteome-wide analysis found no apparent a priori reasons why these 24 proteins would not stimulate CD8 T cell responses, regardless of HCMV serostatus. 

Various experimental approaches could be used to further test this viral evasion hypothesis. Expanding on Sylwester et al. [55], preliminary studies could examine other cohorts not only of adults with chronic infection but also of diverse HLA types, children, or people with primary HCMV infection to understand HLA-, age-, and time-specific T cell responses. To delve into the fine specificity of bulk or sorted T cells for infrequently recognized HCMV proteins, functional assays (e.g., cytokine secretion, cytotoxicity, or proliferation upon stimulation) or non-functional assays (e.g., tetramer or cell surface marker staining) could be conducted. Further, co-culturing T cells with whole-virus-infected, peptide-pulsed, or vector-transfected antigen-presenting or parenchymal cells—with and without intact presentation or other pathways—could explore competition between infrequently (e.g., our Final Set) and commonly detected (e.g., UL83 or UL123) HCMV targets. In addition to investigating MHC or T cell receptor binding, a structural analysis of proteins for epitope locations or modifications, fluorescence-based protein localization within or on the surface of infected cells, and other viral protein or peptide studies might provide insights into their role in evasion. Like early studies of T cell responses to HCMV mutants [87,88] comparative (e.g., transcriptome or proteome) studies of wild-type, engineered, or related viruses might reveal the genetic mechanisms of T cell diversion to “decoy” proteins. Similarly, population and evolutionary genetics analyses might uncover selection pressures that drive viral mutations or T cell specificity. Using combinations of experimental methods and repeated validations, confirming HCMV vulnerability to T cells specific for immunocryptic epitopes could inform the design of an HCMV vaccine that leverages these antigens.

Toward this end, our study illustrates a re-evaluation of past assumptions and strategies as a means to accelerate HCMV vaccine development. We demonstrated that many viral proteins do not generate detectable CD8 T cell responses in healthy adults with chronic HCMV infection yet are predicted to contain multiple epitopes that bind with high affinity to common MHC class I alleles and form pMHC complexes with high immunogenic potential. These data support the hypothesis that, paradoxically, HCMV proteins *infrequently* detected as T cell targets may be effective in stimulating protective HCMV-specific T cells. For a vaccine addressing the global burden of congenital HCMV infection, further proofs-of-concept and experimental data are certainly required to move beyond computational predictions to the practical realm of clinical trials. Since the HCMV proteins identified herein are theoretical candidates, future studies should test these and other predicted immune targets in humans. Perhaps more impactfully, they should also ask whether skewing T cell responses to certain non-essential proteins is an HCMV immune evasion mechanism that shields indispensable proteins from host recognition and, as such, could be exploited as a vulnerability of the virus. Considering non-prototypic viral antigens is but one of many potential answers to Dr. Hanshaw’s momentous call for “*any thoughtful program*” of HCMV vaccine development [1].

## Figures and Tables

**Figure 1 vaccines-11-01629-f001:**
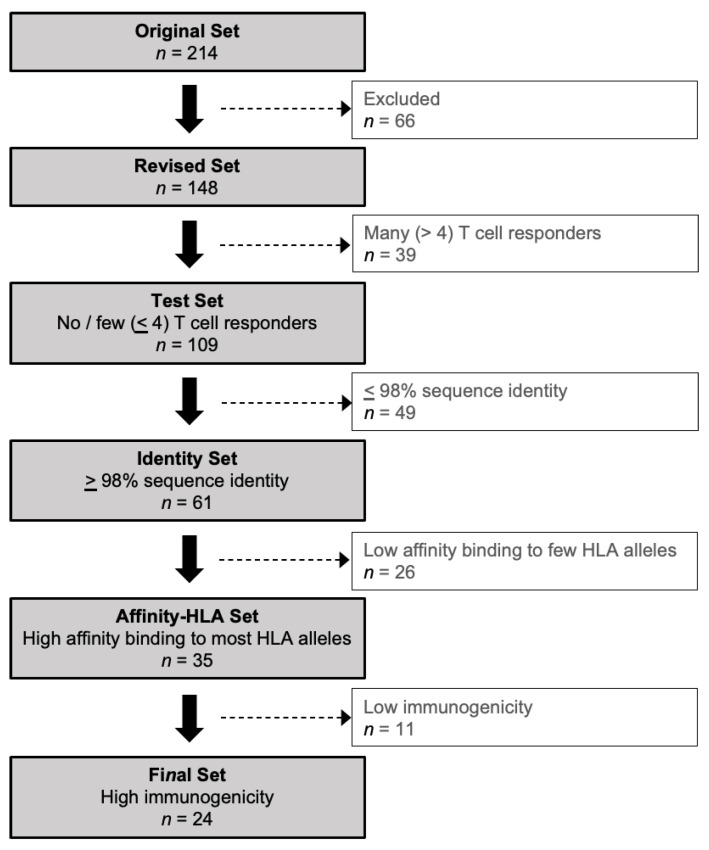
Data analysis algorithm. Tools and thresholds were applied sequentially to yield a Final Set of HCMV proteins predicted to have robust CD8 T cell epitopes. See Methods for description of algorithm and protein sets and Appendix A for details of proteins included and excluded at each step.

**Figure 2 vaccines-11-01629-f002:**
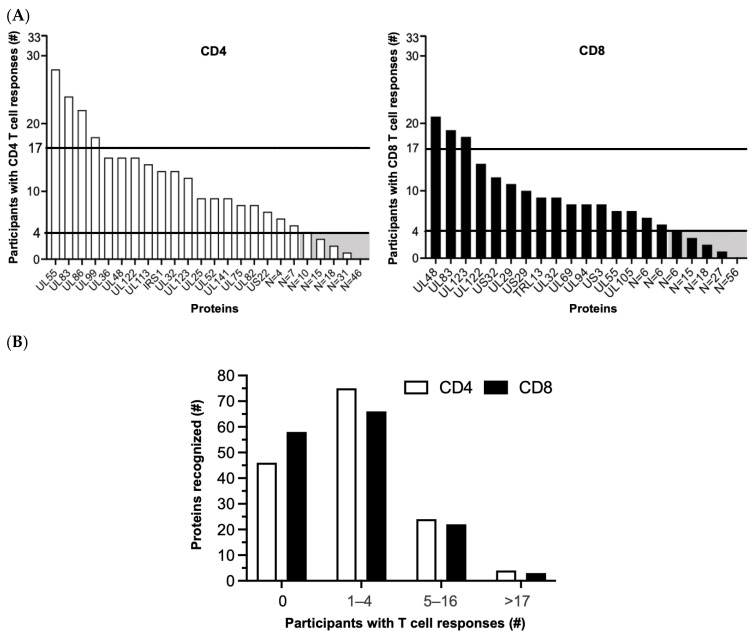
HCMV-specific T cell responses sorted by number of T cell responders. (**A**) Number of participants (*y*-axis) in the original study cohort (*n* = 33; (55)) with detectable CD4 (left panel) or CD8 (right panel) T cell responses against the Revised Set of 148 HCMV proteins (*x*-axis). The far-right end of each *x*-axis notes only the number rather than names of low-frequency proteins. Horizontal lines separate no/few (0–4), some (5–16), and many (≥17) T cell responders. Shaded boxes indicate proteins with no/few responders to which the next step in the algorithm was applied. (**B**) Number of proteins (*y*-axis) recognized by CD4 (white) or CD8 (black) T cells of responders (*x*-axis). Some proteins were recognized by both T cell types. #, number.

**Figure 3 vaccines-11-01629-f003:**
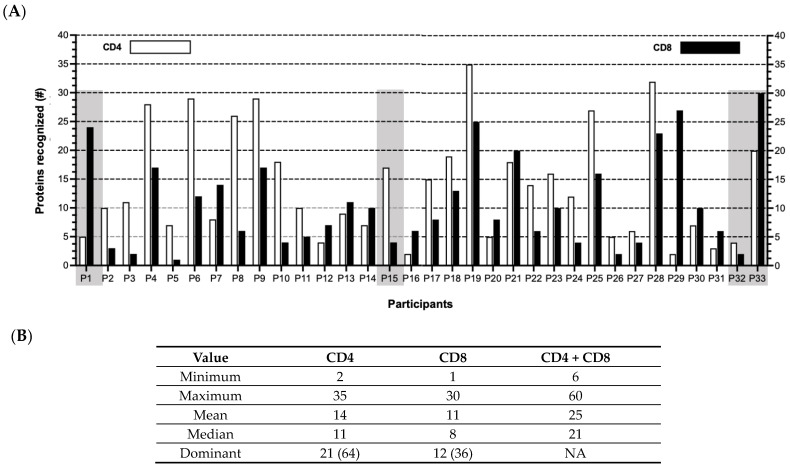
HCMV-specific T cell responses sorted by participants and proteins. (**A**) CD4 (white) or CD8 (black) T cell responses detected in each participant P1–P33 against the Revised Set of 148 HCMV proteins. Shaded boxes highlight examples of participants with predominantly CD8 (P1) or CD4 (P15) or with a wide range (P32 and P33) of responses. (**B**) Number of proteins targeted by CD4 only, CD8 only, or both CD4 and CD8 T cells. Bottom row (“Dominant”) indicates number (%) of participants with more CD4 or more CD8 T cell responses (summary of (**A**)). NA, not applicable.

**Figure 4 vaccines-11-01629-f004:**
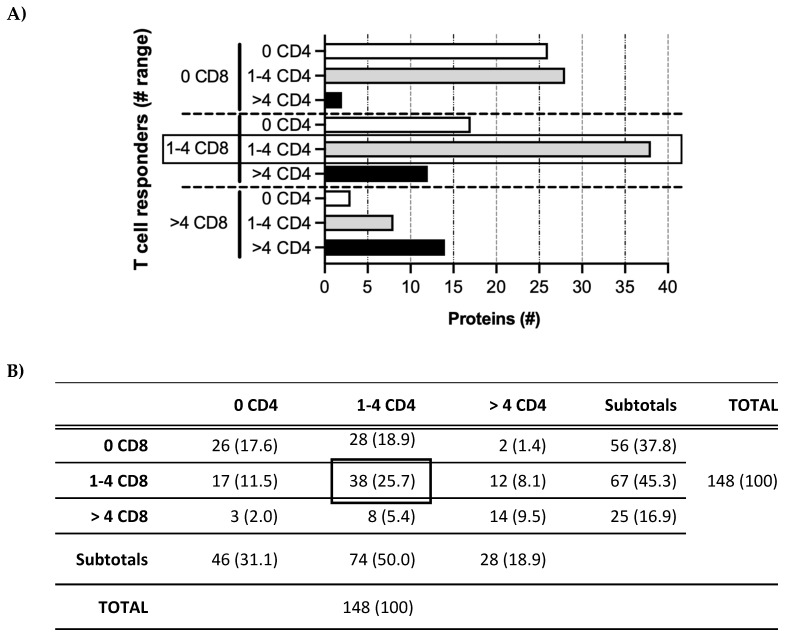
Patterns of T cell responders. (**A**) HCMV proteins in the Revised Set (*x*-axis) recognized by CD4 or CD8 T cells (patterns on *y*-axis) for no (0; white bars), few (1–4; gray bars), or many (>4; black bars) participants. #, number. (**B**) Numbers (percentages) of proteins in each T cell category depicted in (**A**). For example, 38 of 148 (25.7%) HCMV proteins were recognized by few (1–4) CD8 T cell responders and by few (1–4) CD4 T cell responders (boxes in both panels), (summary of (**A**)).

**Figure 5 vaccines-11-01629-f005:**
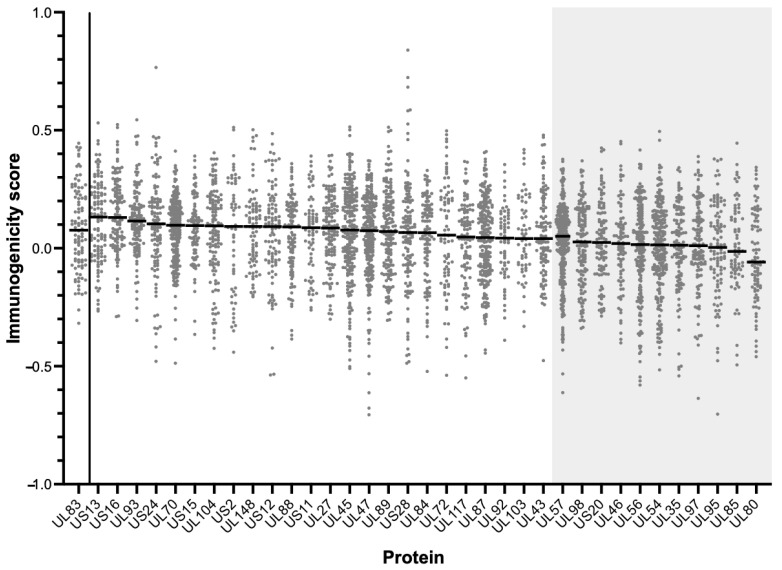
Immunogenicity prediction scores for proteins in the Affinity-HLA Set (*n* = 35) compared to those of UL83 (far left *x*-axis). Shaded areas indicate proteins with median score significantly lower than that of UL83 (Mann-Whitney). Median scores for each protein (solid lines) are noted.

**Table 1 vaccines-11-01629-t001:** Final Set of 24 HCMV proteins (right column) with their values in each protein set of the data analysis algorithm. Test Set columns refer to the number of CD8 or CD4 T cell responders. For comparison, three HCMV proteins commonly reported as T cell targets are shown (gray) but were excluded in the algorithm before the Final Set. See Methods for description of algorithm and protein sets, Figure 1 for overview of algorithm, and Appendix A for details of proteins included and excluded at each step.

Revised Set *n* = 148 Total		Test Set *n* = 109 Total		Identity Set *n* = 61 Total	Affinity-HLA Set *n* = 35 Total		Final Set *n* = 24 Total
HCMV Protein	GenBank or Swiss Prot ID	CD8 (#)	CD4 (#)	Sequence Identity (Median %)	Peptides with IC50 ≤ 50 nM (#)	Restricting HLA Alleles (#)	Immunogenicity Score (Median)
UL 27	CAA35426	0	1	98.7	186	23	0.0860
UL 43	CAA74075	2	4	99.1	151	21	0.0410
UL 45	CAA35404	4	3	99.5	319	26	0.0770
UL 47	CAA35406	2	1	99.7	332	26	0.0740
UL 48	CAA35407	21	15	99.3	542	26	0.0810
UL 70	CAA35386	0	2	99.6	341	25	0.0970
UL 72	CAA35387	0	0	99.0	104	23	0.0550
UL 83 (pp65)	CAA35357	19	24	99.5	137	22	0.0760
UL 84	CAA35358	2	0	98.5	130	23	0.0660
UL 87	CAA35361	3	4	99.6	328	23	0.0450
UL 88	CAA35362	1	0	99.8	148	24	0.0910
UL 89	CAA35363	0	3	99.4	242	24	0.0710
UL 92	CAA35366	0	0	99.5	103	24	0.0430
UL 93	CAA35367	0	0	98.7	172	21	0.1150
UL103	CAA35339	2	0	99.6	93	21	0.0410
UL104	CAA35341	0	0	99.7	170	24	0.0950
UL117	CAA35319	1	0	99.3	121	22	0.0470
UL123 (IE1)	CAA35325	18	12	97.2	115	21	−0.108
UL148Tol	AAA85887.1	0	3	98.7	122	22	0.0920
US 2	CAA35313	2	2	99.5	88	22	0.0920
US 11	CAA35278	1	0	98.6	105	23	0.0870
US 12	CAA35279	3	3	98.9	162	21	0.0910
US 13	CAA35280	0	0	99.6	169	21	0.1320
US 15	CAA35282	0	2	99.6	148	21	0.0960
US 16	CAA35283	2	3	99.4	179	21	0.1300
US 24	CAA35291	3	3	99.6	166	24	0.1030
US 28	P09704	1	0	98.9	208	26	0.0670

## Data Availability

Data is contained within the article or Appendix A.

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
