# Peer review of "Re-Evaluating Human Cytomegalovirus Vaccine Design: Prediction of T Cell Epitopes"

_vaccines, 2023, doi:10.3390/vaccines11111629_

Round 1
Reviewer 1 Report
There are many challenges for design HCMV vaccine. The manuscript described the difficulties on screening protective antigens of HCMV well and analyzed HCMV proteins recognized by CD4 and CD8 T cells, especially CD8 T cells in different populations. Based on the computational predictions, they supposed that HCMV proteins infrequently detected as T cell targets may be protective antigens. The investigation provides some interesting data and opinion, although they are not confirmed by experiments. My comments are as follow:
1. The title is too wide. The authors only analyzed the T cell responses, so it should be specilized in analyzing of T cell responses.
2. It would be better if the authors could do some clinical evaluation to confirm their findings to some extent.
3. There are limitations for the computational methods. The authors should mention their limitations respectively and evaluate their accuracy when these methods are used to do prediction.
Author Response
Reviewer 1
We thank Reviewer 1 for their comments and agree with the concerns expressed. Therefore, we have offered some thoughts in response and edited the manuscript to reflect these helpful points.
- The title is too wide. The authors only analyzed the T cell responses, so it should be specilized in analyzing of T cell responses.
We have changed the title to our preferred version: “Re-evaluating Human Cytomegalovirus Vaccine Design: Prediction of T cell epitopes”. If Reviewer 1 prefers, some alternatives include “Re-evaluating Human Cytomegalovirus Vaccine Design: A Computational Approach to T cell Epitope Prediction” or “Re-evaluating T cell Responses in Human Cytomegalovirus Vaccine Design”. We will await an update on their decision, and if necessary, we or editors can change the title accordingly.
- It would be better if the authors could do some clinical evaluation to confirm their findings to some extent.
We agree that studies with human samples are required to validate T cell epitopes predicted by computational methods.
We note in the Introduction that our study illustrates (rather than validates or proves) one approach to re-evaluating precepts and considering alternative assumptions and strategies for cCMV design: “To illustrate this imperative, we re-examined the question of optimal HCMV epitopes for T cells by considering the hypothesis that HCMV proteins infrequently recognized by T cells may contain targets that favor protection from transmission, infection, or disease. We leveraged computational tools for an in silico rather than experimental approach to T cell antigen discovery across the whole viral proteome.” (lines 119 – 123)
We further explain in the Discussion: “Reverse vaccinology to leverage genomic data [62, 63] is an exciting new process that may inform ex vivo experiments and increase the efficiency and cost-effectiveness of vaccine development, particularly for intractable challenges like HCMV.” (lines 374 – 376) and “For a vaccine addressing the global burden of congenital HCMV infection, further proofs-of-concept and experimental data are certainly required to move beyond computational predictions to the practical realm of clinical trials.” (lines 518 – 539)
Unfortunately, our study had limited scope since we could only perform a computational analysis to test our hypothesis. Further human studies to screen for T cell responses to predicted epitopes in the Final Set would have been the optimal approach, and we hope to perform those studies in the future.
- There are limitations for the computational methods. The authors should mention their limitations respectively and evaluate their accuracy when these methods are used to do prediction.
We welcome this important reminder to include the limitations of the prediction tools. We have now provided more detailed information on prediction tools (lines 185 – 189) and discussed their overall performance (lines 426 – 438, including a new reference that evaluates the accuracy of several tools) was added. While testing the accuracy of the tools was beyond its purpose and scope of this study, the references for each tool we provide contain these analyses.
Reviewer 2 Report
The manuscript describes a study to reevaluate the characteristics of the CD4 and CD8 T cell response to human cytomegalovirus (HCMV). The authors have based their study on previously published data (Sylwester et al, 2005, JEM 202: 673 – 685) that provided an extensive data set of T cell responses in both infected and uninfected individuals. The authors of this study have used modern analytical tools to both refine the set of HCMV proteins to be assessed for potential immunogenicity and to probe HCMV proteins to which there are few responders.
The authors present a logical progression of evaluation to determine the underlying reasons for certain proteins being underrepresented or absent from the responder set. Their findings indicate clearly that lack of predicted immunogenicity does not explain the lack of response. Their conclusion that HCMV may have evolved a mechanism of immune evasion, by carrying proteins that are not essential to the virus but are highly represented in the T cell response, is both intriguing and well supported by their arguments.
Specific Comments:
1. Line 83: the authors state that the co-evolution of HCMV with the human immune system over > 400 million years has enabled the virus to maintain a persistent infection in immunocompetent individuals. While this is a valid explanation, humans have not existed as a species for 400 million years. Therefore, it may be better to state that herpesviruses have co-evolved with the vertebrate immune system for this length of time.
2. The authors provide a strong argument for the use of genomic data for in silico approaches for the design of theoretical HCMV subunit vaccines, based upon predicted immunogenic epitopes that are not represented in the responder set of proteins. A more in-depth discussion of how the prediction would be tested experimentally would be an excellent addition to the manuscript.
Overall, this is an intriguing and well-conceived study that makes some important and testable predictions regarding the immune response to HCMV. The authors have provided a logical argument regarding the experimental approach, and the conclusions suggest a new way of using reverse vaccinology to design vaccines for viral pathogens for which no protective vaccine is currently available.
Author Response
Reviewer 2
We greatly appreciate the supportive comments by Reviewer 2 and were especially delighted that they find this potential viral immune evasion strategy as intriguing as we do. They provided some useful feedback that we incorporated into the revised manuscript.
- Line 83: the authors state that the co-evolution of HCMV with the human immune system over > 400 million years has enabled the virus to maintain a persistent infection in immunocompetent individuals. While this is a valid explanation, humans have not existed as a species for 400 million years. Therefore, it may be better to state that herpesviruses have co-evolved with the vertebrate immune system for this length of time.
This is an important clarification that has been added to the Introduction.
- The authors provide a strong argument for the use of genomic data for in silico approaches for the design of theoretical HCMV subunit vaccines, based upon predicted immunogenic epitopes that are not represented in the responder set of proteins. A more in-depth discussion of how the prediction would be tested experimentally would be an excellent addition to the manuscript.
This suggestion certainly improves the manuscript. Despite the breadth of options, a theoretical approach is offered in lines 481 – 510, including two new references.